# Gene Expression of Mouse Hippocampal Stem Cells Grown in a Galactose-Derived Molecular Gel Compared to In Vivo and Neurospheres

**Keziban Korkmaz Bayram** [1,2,3,*], **Juliette Fitremann** [4], **Arslan Bayram** [5], **Zeynep Yılmaz** [2,6], **Ecmel Mehmetbeyoğlu** [2], **Yusuf Özkul** [2,7] and **Minoo Rassoulzadegan** [2,8]

1 Department of Medical Genetics, Medical Faculty, Ankara Yıldırım Beyazıt University, 06800 Ankara, Turkey
2 Genome and Stem Cell Center (GENKOK), Erciyes University, 38039 Kayseri, Turkey; zeynepyilmaz_55@hotmail.com (Z.Y.); mehmetbeyogluse@cardiff.ac.uk (E.M.); ozkul@erciyes.edu.tr (Y.Ö.); minoo.rassoulzadegan@unice.fr (M.R.)
3 "Central Research Laboratory" Research and Application Center, Ankara Yıldırım Beyazıt University, 06010 Ankara, Turkey
4 Laboratoire des IMRCP, Université de Toulouse, CNRS UMR 5623, Université Toulouse III—Paul Sabatier, 31062 Toulouse, France; fitreman@chimie.ups-tlse.fr
5 Medical Genetics Department of Etlik Zubeyde Hanım Women's Diseases Education and Research Hospital, 06010 Ankara, Turkey; dr.arslan.b@gmail.com
6 Department of Medical Biology, Medical Faculty, Erciyes University, 38039 Kayseri, Turkey
7 Department of Medical Genetics, Medical Faculty, Erciyes University, 38039 Kayseri, Turkey
8 University of Nice, Inserm, CNRS, 06000 Nice, France
* Correspondence: kkorkmazbayram@ybu.edu.tr; Tel.: +90-312-9062-492

**Abstract:** Background: N-heptyl-D-galactonamide (GalC7) is a small synthetic carbohydrate derivative that forms a biocompatible supramolecular hydrogel. In this study, the objective was to analyze more in-depth how neural cells differentiate in contact with GalC7. Method: Direct (ex vivo) cells of the fresh hippocampus and culture (In vitro) of the primary cells were investigated. In vitro, investigation performed under three conditions: on culture in neurospheres for 19 days, on culture in GalC7 gel for 7 days, and on culture in both neurospheres and GalC7 gel. Total RNA was isolated with TRIzol from each group, *Sox8*, *Sox9*, *Sox10*, *Dcx*, and *Neurod1* expression levels were measured by qPCR. Result: *Sox8* and *Sox10*, oligodendrocyte markers, and *Sox9*, an astrocyte marker, were expressed at a much higher level after 7 days of culture in GalC7 hydrogel compared to all other conditions. *Dcx*, a marker of neurogenesis, and *Neurod1*, a marker of neuronal differentiation, were expressed at better levels in the GalC7 gel culture compared to the neurosphere. Conclusions: These results show that the GalC7 hydrogel brings different and interesting conditions for inducing the differentiation and maturation of neural progenitor cells compared with polymer-based scaffolds or cell-only conditions. The differences observed open new perspectives in tissue engineering, induction, and transcript analysis.

**Keywords:** self-assembly; low molecular weight hydrogel; galactolipid; neurosphere; neuron; 3D cell culture; hippocampus; mRNA expression; qPCR

## 1. Introduction

In vitro conditions that reproduce in vivo profile of gene expression is a critical foundation of cell culture. Frustration in this area is high and there is a growing concern for research in particular, given the impact of working on living organisms and on the understanding of the mechanisms involved. Cell culture often reproduces the criteria required very incompletely. However, this approach has obvious practical and financial advantages and has often led to discoveries. Given the complexity of biological research, especially in neuroscience the challenge is topical to find the better conditions in experimental methods

to reveal all aspects of cell complexities. For example, the use of cell culture often leads to stages of differentiation that differ compared to in vivo. Likewise, culture conditions have been shown to produce significant variability.

Currently, new biomaterials are being developed for 3D cell culture techniques to allow the growth and differentiation of neuronal cells. Researchers' greatest expectation for biomaterials is that they can better mimic in vivo behavior. Primary cells, although more difficult to grow than cell lines, exhibit a behavior that is more like cells in vivo. Thus, developing in vitro 3D models starting from primary cells, is likely to provide more relevant results, closer to what is observed in vivo [1–4]. This approach is important notably for gaining a better understanding of the differentiation or interactions between cells or the effect of drugs and so on, in more relevant conditions.

In the case of the central nervous system (CNS), extracts from some specific regions of the brain, notably hippocampus and more especially, embryo hippocampus, contain a high number of neural stem cells and neural progenitor cells. Neurosphere assays have proven to be an efficient method to isolate neural progenitor cells from different parts of the embryonic or adult brain. In neurosphere culture conditions, these cells keep their ability of self-renewal in vitro and upon induction, they can differentiate into neurons, astrocytes, and oligodendrocytes. Therefore, neurospheres are useful in vitro models for mimicking the diversity of cells observed in vivo and their interactions in 3D conditions. Different methods have been described for obtaining neurospheres and the use of this 3D model for different purposes has been discussed [4–6]. Other 3D models relying only on cell self-organization are brain organoids, but growing organoids is more complex compared with neurospheres [7,8].

Alternatives to cell-only 3D models introduce scaffolds made of biomaterials, which intend to mimic the extracellular matrix function. The interactions between the cells and the scaffold strongly affect the cell proliferation, 3D distribution in space, differentiation, cell–cell interactions, and so on. Hence the scaffold cannot be considered just as a neutral support for the cell growth in 3D. Highlighting the impact of the scaffold on the gene expression of the cultivated cells is thus very important in order to better select the right scaffold for a given application [9,10]. Many biomaterials of different types are currently being developed for addressing in vitro 3D cell cultures of neuronal cells. They are most often derived from the extracellular matrix (ECM) or its components [9–11] or made of bio sourced polymers [9,12–14] and purely synthetic [15] or bio sourced/synthetic composites [11,16]. Emerging alternative biomaterials for neuron cell culture are self-assembling fibrillar scaffolds [17–26]. In these materials, the formation of the gel is not based on a polymer network but results from the self-assembly of small molecules of low molecular weight through non-covalent interactions into a "self-assembling fibrillar network" (SAFIN). Interestingly, these gels are very soft at the macroscopic and microscopic scale, which is favorable to the growth and differentiation into neurons [20,27,28]. Secondly, they form fibers with very various sizes, morphology, and local stiffness that are gelator-specific and may provide original mechanical, geometrical, and topographical cues for cells. In particular, it has been shown that the morphology of the fibers at the nanoscale, including helix chirality or nanogrooves impacts the cell behavior [20,29,30]. In the case of neurons, the differentiation of a neurite into an axon or dendrite depends on the rigidity of the substrate [28], and the path taken by a neurite depends on mechanical signals [31]. Mechanical and topological signals may be quite original in the case of molecular hydrogel scaffolds and may induce significant differences in neural cell cultures.

The objective of this work was to explore the difference in cell expression profile of primary cells extracted from the hippocampi of mouse embryo brains, in four different conditions (Figure 1): (1) in vivo at day 19.5 pc; (2) grown in N-heptyl-D-galactonamide supramolecular hydrogel (GalC7) for 7 days; (3) grown in neurospheres (pHEMA) for 19 days; and (4) grown in neurospheres (pHEMA) for 12 days then in N-heptyl-D-galactonamide supramolecular hydrogel (GalC7) for 7 days. The neurospheres are formed in non-adhesive conditions, in culture plates coated with poly(2-

hydroxymethacrylate). N-heptyl-D-galactonamide is a synthetic carbohydrate-based low molecular weight gelator that self-assemble in a network of wide ribbons supporting the supramolecular hydrogel (Figure 2). Since it is a small synthetic molecule and not a polymer, it is not submitted to batches variations. Therefore, more reproducible conditions for cell culture are expected. In a previous study, it has been shown that N-heptyl-D-galactonamide supramolecular hydrogel is suitable for the growth and differentiation of adult human neural stem cell (hNSC) in 3D [20]. Additionally, 3D constructs can be made by 3D printing [32,33]. However, this hydrogel is nearly completely consumed by the cells after 7 days. For this reason, we introduced in the study design the condition (4) in which a 19-days old culture is performed, but with the final 7 days carried out on the GalC7 hydrogel. Thus, the conditions (3) and (4) will highlight how the abundance of neural progenitor cells is affected by early-stage cell culture conditions and also how the GalC7 hydrogel can change the response of a 12 days neurosphere culture.

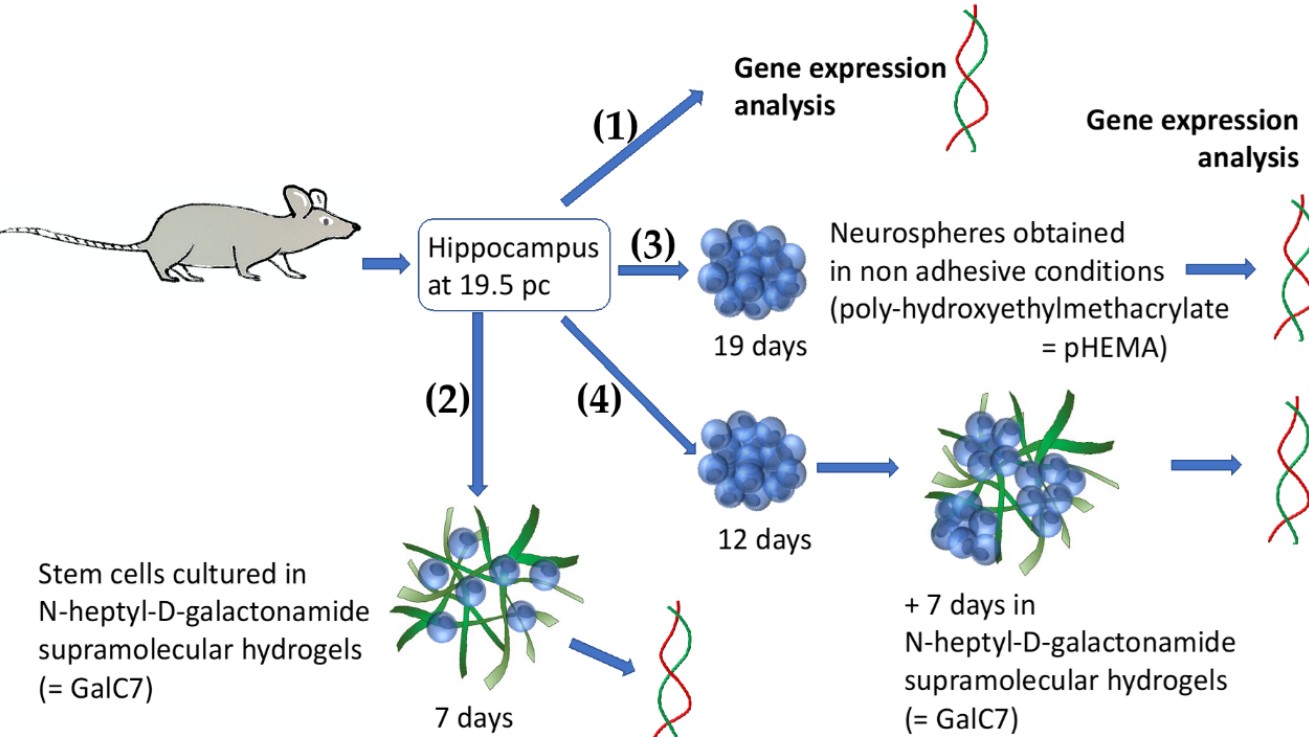

**Figure 1.** Study design: four conditions for the cell culture of mouse embryo hippocampi primary cells followed by gene expression analysis by qPCR. (**1**) in vivo at day 19.5 pc; (**2**) grown in N-heptyl-D-galactonamide supramolecular hydrogel (GalC7) for 7 days; (**3**) grown in neurospheres (pHEMA) for 19 days; and (**4**) grown in neurospheres (pHEMA) for 12 days then in N-heptyl-D-galactonamide supramolecular hydrogel (GalC7) for 7 days.

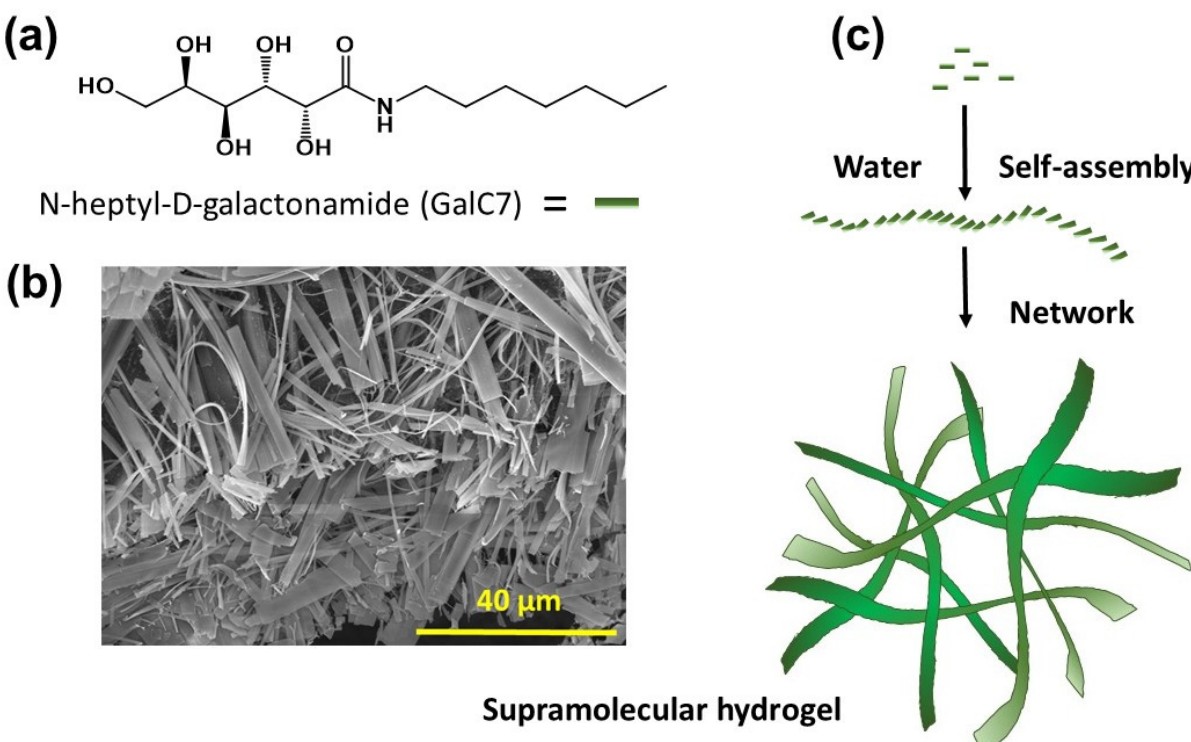

**Figure 2.** (**a**) Molecular structure of N-heptyl-D-galactonamide. (**b**) Hydrogel fibers (cryo-SEM) and (**c**) mechanism of the supramolecular hydrogel formation.

For studying the cell expression profile, five genes were selected to highlight the differentiation of the cells either in astrocytes, oligodendrocytes, or neurons. Sex Determining Region Y (SRY)-box 8 (*Sox8*), SRY-box 9 (*Sox9*), and SRY-box 10 (*Sox10*) are three genes coding for the three transcription factors Sox8, Sox9, and Sox10 that form the subgroup SoxE of Sox transcription factors. In vivo, they regulate the gene expression in different tissues and at different times of development or adult life [34–37]. In the context of the CNS, Sox9 has a key role in neural stem cell induction and survival and for their differentiation into glial cells, either astrocytes or oligodendrocytes. During embryogenesis, it is expressed by astrocyte precursor cells and at a lower level, by oligodendrocyte precursor cells [37–39]. Then Sox9 continues to be expressed at a high level in mature astrocytes while it stops in mature oligodendrocytes. As a result, Sox9 is often used as a specific marker of mature astrocytes [39–48]. Conversely, the Sox10 transcription factor is expressed at a high level in mature oligodendrocytes and is essential for the myelinization process, while Sox10 expression is turned off in mature astrocytes and early in neuron lineage [37]. Thus Sox10 is generally considered as a marker of oligodendrocytes [38], but it is also expressed by immature glial cells [49]. Sox8 also acts in these processes. Despite a role often redundant with Sox10 and Sox9 factors and less critical than Sox9 and Sox10, it is coexpressed with those factors at different stages. Notably, Sox8 function becomes prominent late in myelinated oligodendrocytes and interacts with Sox10 for myelin maintenance [34,44,48,50].

Two markers of neurons have been selected. Doublecortin (Dcx) is a marker transiently expressed by neuronal precursor cells in vivo [51,52]. It is essential for the normal development of the hippocampus and lamination. It is expressed nearly exclusively in dividing immature neurons and is not expressed by the glial progenitor cells. Thus, it is considered a marker of neurogenesis [53–56]. *Neurod1* is a transcription factor acting in the conversion of neural progenitor cells to mature neurons in vivo. It is considered a neuronal differentiation factor. It is highly expressed during the development of neurons in the peripheral and CNS. It enables the survival, migration, and maturation of newborn neurons in the brain during development and adulthood [57,58]. It is also essential for the normal development of the hippocampus [58–60]. It allows the maintenance and repair of

tissue [61] and it has been shown also that it can be used for reprogramming glial cells to neurons and for reversing glial scar [62].

In this study, the quantitative analysis of the gene expression of these five markers in mouse embryo hippocampi primary cells grown in the four conditions described in Figure 1 was made by qPCR. The results are detailed below.

## 2. Material and Methods

N-Heptyl-D-galactonamide, which can be purchased from Innov'Orga (Reims, France), was synthesized according to the protocol described in the work of Chalard, A. et al. [20].

### 2.1. Mice

Eight-week-old *Balb/c* inbred mice were used. The mice were housed at a room temperature of 20–24 °C and a relative humidity of 45–70% in cages with a minimum area of 180 cm$^2$ and a minimum height of 12 cm. Once the mice reached sexual maturity, they were mated (one male and two females). In the early morning, the mice were checked for the presence of a vaginal plug, a marker of mating in females. Mice with vaginal plugs were considered as pregnant for 0.5 days. After 19 days of pregnancy (E19.5), the mice were sacrificed, and the uterus removed into phosphate-buffered saline (PBS). Embryos were removed from the uteri of pregnant mice on E19.5 with forceps. In total four male embryos were used in this study: one for each group/experiment. The protocols were approved by Erciyes University Animal Ethics Committee, number 16/132.

### 2.2. Dissection of Embryonic Hippocampus

After the whole brain of each embryo was removed, it was collected on a slide and incubated for 2 min at −20 °C. The right and left hippocampi were aspirated from 4 different embryos with an individual pipette tip and put into different Eppendorf tubes. For condition 1 aspirated fresh embryonic hippocampi of first embryo used for RNA isolation, 300 µL TriPure Isolation Reagent (Roche, Germany) was added to the aspirated hippocampus. For condition 2, 3, and 4 aspirated embryonic hippocampi of other three embryos washed with 1 mL Hanks' balanced salt (HBS) solution and cultured. The second embryo was used for condition 2: primary cells from the embryonic hippocampus cultured on GalC7 gel for 7 days, the third embryo was used for condition 3: primary cells cultured as neurospheres for 19 days, and the fourth embryo was used for condition 4: cells grown as neurospheres for 12 days, then grown 7 days further on GalC7. Tissue was taken from the tail of each embryo, and DNA was isolated for gender determination (see the "Gender Detection of E19.5 Embryos" section).

### 2.3. Gender Detection of E19.5 Embryos

a.    DNA isolation

Tissue fragments from the embryos were transferred to Eppendorf tubes containing 300 µL of lysis buffer (20 mM Tris-HCl PH 8, 100 mM EDTA PH 8, 0.1% sodium dodecyl sulfate (SDS), distilled water, and proteinase K 400 µL/mL). The samples incubated overnight at 55 °C. After incubation, 100 µL of ammonium acetate was added and mixed thoroughly. The samples were centrifuged at 4000 rpm for 20 min. The supernatants were transferred to clean Eppendorf tubes, and 500 µL of isopropanol was added and incubated for 5 min at room temperature. After incubation, centrifugation was performed for 10 min at 4000 rpm, and 1 mL of 70% ethanol was added to the pellet that remained after the supernatant was discarded. The samples were again centrifuged at 4000 rpm for 10 min, and the supernatant was discarded. The Eppendorf tubes were incubated with the lids open for 10 min at room temperature to remove the alcohol from the resulting pellets. After drying, the pellets were dissolved in 100 µL of nuclease-free water (NFW) (Qiagen, Germany). The DNA concentration was measured using a Biospec Nano spectrophotometer (Shimadzu, Kyoto, Japan).

b. Polymerase chain reaction (PCR)

After DNA isolation, *Sry* forward (5′TGCACAATTGTCTAGAGAGC3′) and reverse (5′ACTGCAGAAGGTTGTACAGT3′) and *Pax6* forward (5′CTTTCTCCAGAGCCTCAAT3′) and reverse (5′GCAACAGGAAGGAGGGGGAGA3′) primers were used for PCR amplification (Table 1). Five microliters DNA (10 pg–1 μg), 5 μL 10X Taq Buffer, 4 μL 25 mM MgCl$_2$, 4 μL dNTP mix (2.5 mM each), 0.5 μL 10 μM each primer, 0.5 μL 1.25 U Taq DNA Polymerase (Thermo Fisher, Waltham, MA, USA), and water (depending on the reaction volume) were mixed in a 0.2 mL tube. The samples were incubated in a thermocycler (SensoQuest, Göttingen, Germany) under the following conditions: predenaturation at 95 °C for 5 min, 1 min of denaturation at 94 °C for 30 cycles, annealing at 57 °C for 45 s, extension at 72 °C for 1 min, and finally 10 min at 72 °C. The PCR products were run on a 2% agarose gel containing ethidium bromide. The gel was imaged using a gel imaging system (Bio-Rad, Hercules, CA, USA) and the Image Lab analysis software (Figure 3).

**Table 1.** Primer sequences used for gender determination.

| Target Gene | Primer Sequences (5′–3′) | Product Size (bp) | Primer Concentration (μM) |
|---|---|---|---|
| *Sry* | Forward: TGCACAATTGTC-TAGAGAGC Reverse: ACTGCAGAAG-GTTGTACAGT | 329 | 10 μM |
| *Pax6* | Forward: CTTTCTCCAGAGC-CTCAAT Reverse: GCAACAGGAAG-GAGGGGGAGA | 150 | 10 μM |

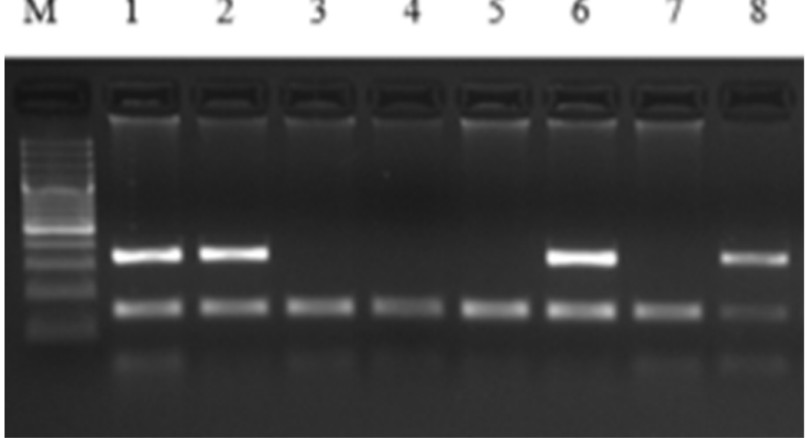

**Figure 3.** Gender determination gel image. M indicates marker (100–3000 bp). The 1st, 2nd, and 6th embryos were males, and the 3rd, 4th, and 5th embryos were females. The 7th sample was a positive control female, and the 8th sample was a positive control male.

### 2.4. Preparation of the Hydrogel in Culture Plates

For each sample, 9 mg of GalC7 was weighed and poured into a flask and 2 mL of ultrapure water (Merck, Kenilworth, NJ, USA) was added. The volume of the flask must be at least five times greater than the volume of water introduced (typically 10 mL, to avoid a too high pressure in the flask when heated). A programmable oven was heated to 115 °C, and the flask was placed in the oven with the lid slightly closed. GalC7 in powder form

was incubated intermittently in the oven until it dissolved. When the powdered GalC7 was dissolved, the temperature of the oven was reduced to 105 °C, and 24-well culture plates were placed in the oven. On the outside, distilled water inside of a metal tray (large enough to fit the plate) was put on the incubator. When distilled water boiled, 500 µL of GalC7 was pipetted gently but be very quickly to avoid the setting of the gel into the pipette cone.

Before the GalC7 in the plate became a gel, the plate was quickly put on the oven and sandwiched between two empty plates. A steel block was placed on the top of the plates, and the temperature of the oven was slowly decreased from 105 °C to room temperature (minimum of 90 min). Thus, the gel was formed by slow cooling to protect it from thermal shock, providing a more homogeneous gel. When the gel was formed, the plate was moved to a laminar cabinet, where the lid was opened and the water droplets on the well walls were removed with a pipette. Then, 300 µL of medium was pipetted very slowly onto the gel and incubated at 37 °C for 5 h in an oven containing 5% $CO_2$. After incubation, the old medium was withdrawn with a pipette, and 300 µL of medium was added again and incubated overnight at 37 °C in an oven containing 5% $CO_2$. After incubation, the old medium was withdrawn with a pipette, and 300 µL of medium was added and incubated again in an oven containing 5% $CO_2$ at 37 °C for 5 h. The aim was that the water in the gel was replaced with the medium. The medium was not changed during the 7 days of culture.

### 2.5. Primary Cell Culture

a.    Primary cell culture in culture plates coated with poly(HEMA)

The following procedures were performed for primary culture of cells from the hippocampus: Poly(hydroxyethylmethacrylate) (pHEMA) (Sigma, Darmstadt, Germany) was prepared one day before the start of culture. A total of 300 mg pHEMA was dissolved in 10 mL of 95% ethanol in a water bath (65 °C) with occasional vortexing. pHEMA (200 µL) was pipetted into each well of a 24-well plate in a laminar cabinet. The plate was left in the cabinet overnight with the lid open to allow the alcohol to evaporate. The following day, the right and left hippocampi were aspirated from the embryos of pregnant mice at E19.5 and were taken up in 1 mL HBS solution. Tissues were resuspended and centrifuged at 1000 rpm for 2 min at room temperature. Dulbecco's modified Eagle medium/nutrient mixture F-12 (DMEM/F12) (Sigma Germany) was used for the cultivation of cells. Fetal bovine serum (FBS; 10%) (Sigma, Germany), 1% penicillin-streptomycin (pen-strep) (Gibco, Life Technologies, Carlsbad, CA, USA), and L-glutamine (200 mM) (Gibco, Life Technologies) were added to the DMEM/F12, and ready-to-use medium was obtained. After centrifugation, the supernatant was discarded, and the cells were resuspended in 1 mL ready-to-use medium. Each sample was seeded in two pHEMA-coated plates (500 µL per well). The plates were incubated at 37 °C in an oven containing 5% $CO_2$. On the 3rd day of culture, 1X Insulin-Transferrin-Selenium-A (ITS) and 1X B27 supplement were added to each well. Step 7 was repeated on the 7th day of culture. On the 12th day of culture, the neurospheres were evenly distributed into two separate Eppendorf tubes using a mouth pipette under an inverted microscope. The neurospheres from one Eppendorf tube was cultured with GalC7 for 7 days, and the others were transferred to a new pHEMA coated plate and cultured for 7 days more.

b.    Primary cell culture with GalC7 gel

Aspirated embryonic hippocampal tissues, were washed with HBS and then centrifuged at 1000 rpm for 2 min at room temperature, and the pellet was resuspended in 250 µL of ready-to-use medium. 1X B27 supplement and 1X ITS were added to the samples, and the samples were gently inverted. The resuspended samples were carefully pipetted into wells containing GalC7 gel to not damage the gel. The plate was incubated at 5% $CO_2$ at 37 °C. Seven days later, the culture was terminated, and RNA was isolated (see method below).

c.    Cell culture of neurospheres with the GalC7 gel

When the cultures were 12 days old, the neurospheres were transferred to Falcon tubes, and 500 µL of 0.25% trypsin-EDTA (Gibco, Life Technologies) was added to the neurospheres for a total volume of 2 mL and incubated at 37 °C for 5 min. After the addition of 1 mL of ready-to-use medium to the Falcon tubes, the tubes were centrifuged at 1000 rpm for 2 min at room temperature. After the supernatants were discarded, the pellets were resuspended with 250 µL of ready-to-use medium. After the addition of 1X B27 supplement and 1X ITS to the samples, the samples were cultured in wells containing GalC7 gel (Figure 4). The culture was stopped after 7 days, and RNA was isolated.

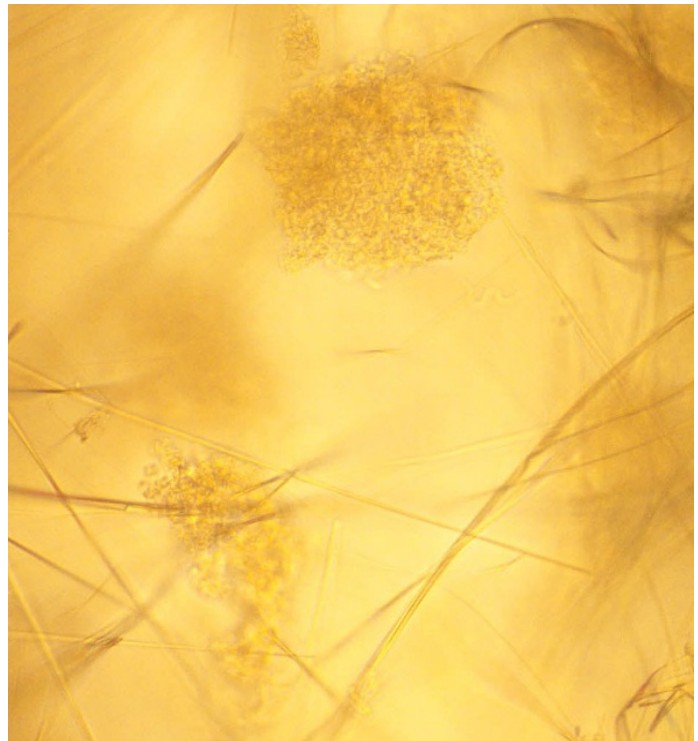

**Figure 4.** Image of 3D neurosphere culture in GalC7 gel (magnification at 20×).

*2.6. RNA Isolation*

RNA was isolated from male embryos identified by PCR.

a.      Isolation of RNA from E19.5 embryonic hippocampi

Three hundred microliters TriPure Isolation Reagent (Roche, Mannheim, Germany) was added to hippocampal tissues, and the tissues were homogenized with a syringe. One hundred microliters of chloroform was added, and the samples were vortexed for 15 s. The tubes were centrifuged at 10,000 rpm for 20 min at 4 °C. After centrifugation, the aqueous phase was transferred to a new Eppendorf tube. Three hundred microliters isopropanol was added to the tube, and the aqueous phase was collected, vortexed well, and stored at −20 °C overnight. After incubation, the tubes were centrifuged at 10,000 rpm for 10 min at 4 °C, and the supernatant was discarded. One milliliter ethanol was added to the pellet for washing, and the tubes were gently inverted a few times. The tubes were centrifuged at 8000 rpm for 5 min at 4 °C, and the supernatant was discarded. The washing process (Step 7) was repeated, and the supernatant was discarded after centrifugation. The pellet was dissolved in 30 µL NFW (Qiagen, Hilden, Germany), and absorbance measurements were performed using a Biospec Nano spectrophotometer (Shimadzu, Kyoto, Japan).

b.      Isolation of RNA from primary cells cultured with pHEMA

When the cultures were 19 days old, the contents of the pHEMA-coated plate were transferred to Eppendorf tubes with Pasteur pipettes. At the end of the culture, the following protocol was used for RNA isolation. After the tubes were centrifuged at 1000 rpm

for 2 min at room temperature, 1 mL of TriPure Isolation Reagent (Roche, Germany) and 1 μL (20 ng/mL) of glycogen were added, and the tubes were vortexed thoroughly. Two hundred microliters of chloroform was added to the tubes, and they were vortexed for 15 s. The tubes were centrifuged at 10,000 rpm for 20 min at 4 °C. After centrifugation, the aqueous phase was transferred to a new Eppendorf tube. Then, 500 μL of isopropanol was added to the tube, and the aqueous phase was collected, vortexed well, and stored at −20 °C overnight. After incubation, the tubes were centrifuged at 10,000 rpm for 10 min at 4 °C, and the supernatant was discarded. One milliliter of ethanol was added to the pellet for washing, and the tubes were gently inverted a few times. The tubes were centrifuged at 8000 rpm for 5 min at 4 °C, and the supernatant was discarded. The washing process was repeated, and the supernatant was discarded after centrifugation. The pellet was dissolved in 30 μL of nuclease-free water (Qiagen, Germany), and absorbance measurements were performed using a Biospec Nano spectrophotometer (Shimadzu).

c.　　Isolation of RNA from primary cells cultured with GalC7 gel

At 7 days of culture, the medium was withdrawn carefully as to avoid damaging the gel. Two hundred microliters of PBS was pipetted onto the gel and withdrawn to remove any non-adherent cells from the gel. The following procedures for RNA isolation were then performed. One milliliter of TriPure Isolation Reagent (Roche, Germany) was added to each well containing GalC7 gel by pipetting to dissolve the gel. TriPure Isolation Reagent and the cell mixture were transferred from the wells to Eppendorf tubes by pipetting. The tubes were thoroughly vortexed, and then 1 μL (20 ng/mL) of glycogen and 200 μL of chloroform were added to the Eppendorf tubes, which were vortexed for 15 s. The tubes were centrifuged at 10,000 rpm for 20 min at 4 °C. After centrifugation, the aqueous phase was transferred to a new Eppendorf tube. Then, 500 μL isopropanol was added to the tubes, and the aqueous phase was collected, vortexed well, and stored at −20 °C overnight. After incubation, the tubes were centrifuged at 10,000 rpm for 10 min at 4 °C, and the supernatant was discarded. One milliliter of ethanol was added to the pellet for washing, and the tubes were gently inverted a few times. The tubes were centrifuged at 8000 rpm for 5 min at 4 °C, and the supernatant was discarded. The washing process was repeated, and the supernatant was discarded after centrifugation. The pellet was dissolved in 30 μL NFW (Qiagen, Germany), and absorbance measurements were performed using a Biospec Nano spectrophotometer (Shimadzu).

### 2.7. Complementary DNA (cDNA) Synthesis

The EvoScript cDNA Kit (Roche, Germany) was used to synthesize cDNA from RNA samples. A mixture of 4 μL reaction buffer (5X) and 14 μL template RNA (2.5 μg/20 μL) was kept on ice for 5 min. At the end of this period, 2 μL enzyme mixture was added to each sample, and the samples were incubated for 15 min at 42 °C, 5 min at 85 °C and 15 min at 65 °C in a thermocycler (SensoQuest) (Table 2). The cDNA samples obtained after incubation were used for preamplification.

**Table 2.** Conditions for cDNA synthesis.

| | |
|---|---|
| Reaction buffer (5X) | 4 μL |
| Template RNA (2.5 μg/20 μL) | * |
| NFW | ** |
| kept on ice for 5 min. | |
| Enzyme mixture | 2 μL |
| 5 min at 42 °C<br>5 min at 85 °C<br>15 min at 65 °C | |

* Hippocampus, 2.5 μL; neurosphere, 7 μL; Gal-C7 gel, 7 μL. ** depending on the volume of template RNA.

### 2.8. Preamplification Method

The Pre-AMP Master Kit (Roche, Germany) was used for preamplification. Primers/Probes (Integrated DNA Technologies, Belgium) were diluted by 10%. For preamplification, 5 µL of cDNA, 10 µL of Preamp master mix, 1.4 µL of diluted primers/probes, and nuclease-free water that complete the final volume to 25 µL were used. Sample tubes were incubated at Thermal Cycler (SensoQuest) for 1 min at 95 °C for 15 min at 95 °C for 15 cycles at 4 °C for 60 min.

### 2.9. Measurement of mRNA Expression Levels by qPCR

To measure mRNA expression levels, a mixture of 10 µL of 2X Probe Master Mix (Roche, Germany), 1 µL of primer/probe (Integrated DNA Technologies, Leuven, Belgium), and 4 µL of nuclease-free water was pipetted into 96-well plates (Roche, Germany). Five microliters of preamp cDNA (diluted 1/40) was added to each well on the plate. Whole samples were studied in duplicate. The well plates were sealed and placed in a LightCycler 480 II (Roche, Germany) for 10 min at 95 °C, 10 s at 95 °C for 50 cycles, 30 s at 60 °C, 1 s at 72 °C, and 30 s at 40 °C for cooling. *Beta-actin* was used as a housekeeping gene. The data were analyzed using the 2DD Ct method and normalized using the values obtained in the ex vivo conditions "hippocampi 19.5 pc" as the calibrator.

## 3. Results and Discussion

### 3.1. Culture and Conditions

Four conditions for primary hippocampal cells were chosen following schematic presentation (Figure 1). Once steady culture conditions were achieved, total RNA was extracted at different time points (see the "Materials and Methods" section for culture conditions). To extract total RNA from hippocampal cells in culture under different conditions, including GalC7 gel, we introduced modifications to the routine Trizol method. The GalC7 gel is soft and is easily damaged during washing steps. To avoid breaking the gel during the washing steps, we lysed the cells together with the gel. We added TriPure Isolation Reagent to cell-free gel, which was perfectly lysed under these conditions, and we checked its specific absorbance and interference with qPCR. Absorbance measurements were performed by applying an RNA isolation protocol to an empty gel. We observed that the peak of the GalC7 gel (Figure 4) was very similar to the RNA peak we previously observed. Thus, we used an RNA sample with a known Ct value of a housekeeping gene (*beta-actin*) and mixed the RNA sample with the product isolated from the cell-free GalC7 gel. Then, the Ct value of this mixture was determined, and we found that the *beta-actin* Ct value was unchanged. These results demonstrated that GalC7 gel did not inhibit PCR. After this experiment, total RNA was isolated from GalC7 gel containing embryonic hippocampal neurons (see profile of RNA in Figure 4).

### 3.2. Gene Expression Studies with qPCR

In Figure 5 is reported the transcript expression analyzed from cells grown in the following conditions: (condition 1): cells obtained from fresh embryonic hippocampi E19.5 pc.; (condition 2): primary cells from the embryonic hippocampus cultured on GalC7 gel for 7 days, (condition 3): primary cells cultured as neurospheres for 19 days, and (condition 4): cells grown as neurospheres for 12 days, then grown 7 days further on GalC7.

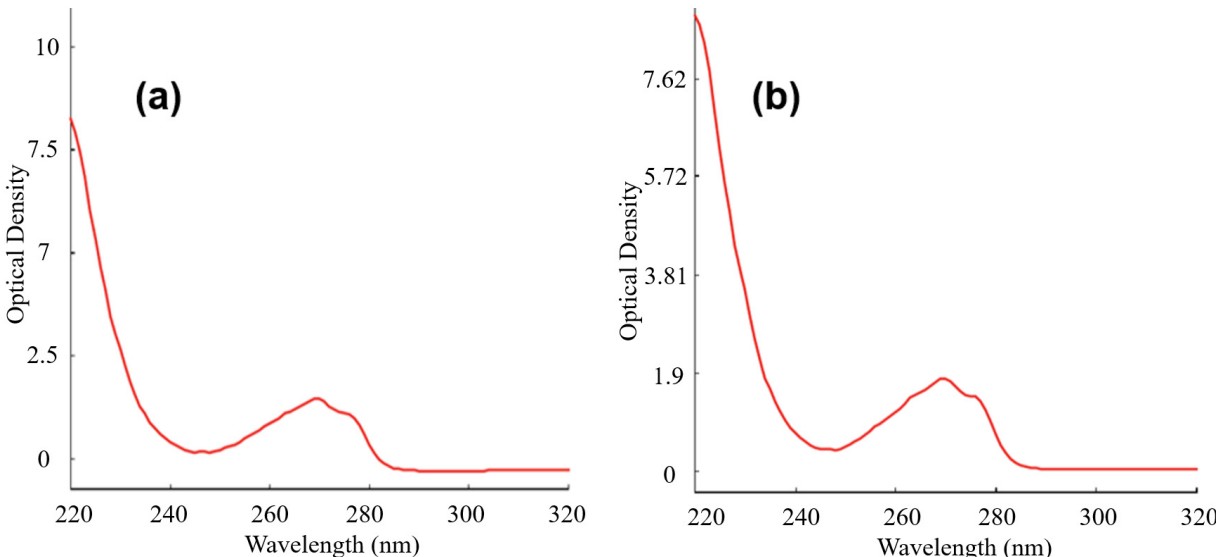

**Figure 5.** The absorbance value of GalC7 gel without (**a**) and with (**b**) cultured neuronal cells.

The transcript levels of *Sox8, Sox9*, and *Sox10* were found to be much higher in cells cultured on GalC7 gel after 7 days than in cells from the other groups (Figure 5, *Sox8*, *Sox9*, and *Sox10*). Based on the results, it can be concluded that the culture on GalC7 hydrogel directs the fate of the hippocampal primary cells predominantly towards glial cells, either oligodendrocytes and/or astrocytes, compared with in vivo (condition 1). When the cells are grown as neurospheres for a longer time (19 days) (condition 3) *Sox 8* and *Sox 10* are not expressed anymore, which show that astrocytes would be the main population of cells in this condition. Growing the neurospheres during only 12 days and then for 7 days more in GalC7 hydrogel does not change this profile. It shows that in the case of the glial cells, the fate of the cells and the resulting transcripts have been determined at an early stage of the cell culture by the conditions on which the cells are grown and cannot be changed by adding GalC7.

Otherwise, *Dcx* and *Neurod1* transcript levels were much higher in fresh embryonic hippocampi than in other conditions (Figure 6, *Dcx* and *Neurod1*). It is consistent with the fact that a high level of neurogenesis and neuronal differentiation is expected in hippocampi. Conversely, in in vitro conditions, cells grown as neurospheres for 19 days did not express these markers. Otherwise, cells grown in GalC7 hydrogels can express at some level the two markers at day 7, around 1/4 the one observed in ex vivo for *Neurod1*. Even if the transcription level of *Dcx* remains lower than in the hippocampi, the presence of *Dcx* transcripts after 7 days of culture on GalC7 gel suggests the presence of neurogenesis in these conditions. The cell culture was not performed longer on GalC7 because the gel was consumed by the cells within 7–10 days, thus gene expression on a longer time could not be quantified. However, from the results of condition 4 (neurospheres for 12 days, then GalC7 for 7 days), it is observed that the level of *Neurod1* was better maintained when the neurospheres are grown for 7 extra days on GalC7 gel instead of being grown in the neurospheres culture condition only (condition 3, 19 days). The presence of a relatively high level of *Neurod1* transcripts after 7 days of culture on GalC7 gel suggests the differentiation of some cells into neurons in condition 2 and in condition 4. It is worth noting that these results were obtained without the addition of any extra factors, the GalC7 hydrogel being the only extra molecule added to the culture conditions.

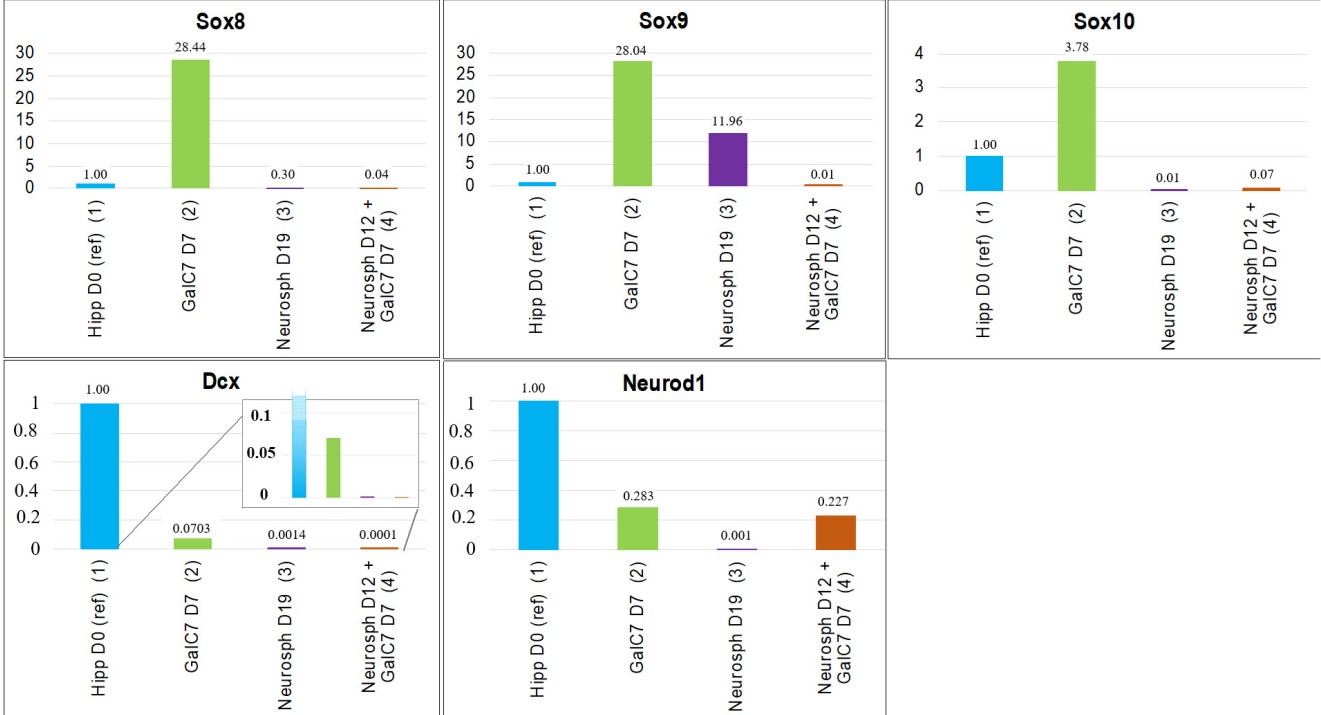

**Figure 6.** Transcript analysis qPCR results. (**1**) Hippocampi at day 19.5 pc. (**2**) Cells cultured on GalC7 gel for 7 days (D7), (**3**) cells cultured as neurospheres with pHEMA coatings for 19 days (D19), and (**4**) cells cultured as neurospheres with pHEMA coatings for 12 days (D12) then on GalC7 gel for 7 days (D7).

The results raise the question why this hydrogel would favor glial cells and possibly oligodendrocytes, and to a lesser extent, neurons. N-heptyl-D-galactonamide is an acyclic galactose derivative, in which the anomeric carbon of the galactose is oxidized and linked with an amide bond to a fatty chain. It means that N-heptyl-D-galactonamide fibers display molecular patterns, a polyol configuration, that could mimic at some points galactolipids. Galactocerebrosides (GalC) are glycolipids produced by oligodendrocytes at a high level. They are also highly specific of those cells so that immunolabeling targeting galactocerebrosides (anti-GalC) is used for the identification and separation of oligodendrocytes [63].

These glycolipids play an important role in oligodendrocyte differentiation, maturation, myelinization, and oligodendrocyte-axon interaction [64,65]. Up to now, preliminary studies on this scaffold [37] had focused on the effect of the microscopic architecture and mechanical properties on the nervous cell survival and early differentiation markers. It is possible also that the chemical structure of the molecule might have influenced the fate of the cells. The high level of *Sox10* factor might be due either to the early stage of the culture (7 days), because both *Sox8*, *Sox9*, and *Sox10* are expressed in immature glial cells or might highlight an effect of GalC7 favoring oligodendrocytes.

The results tend to indicate also that the GalC7 hydrogel is a favorable in vitro condition for neurogenesis and the differentiation of the primary cells into neurons. The fibrillar network provides mechanical signals and topological clues that are very different compared with the neurospheres culture conditions. In the case of GalC7, the fibrillar network is made of coarse heterogeneous ribbons, some of them being very large, at least the size of a cell. The electronic microscopy (Figure 2) of these fibers suggest that they have nanometric grooves along the fibers, that may have a strong impact in the development of cell projections, especially neurites, which are known to be mechanosensitive and sensitive to topography. It might explain the higher expression of *Neurod1* transcript when neurospheres at day 12 are contacted secondary during 7 days with GalC7 (condition 4, 19 days), compared with neurospheres only condition (condition 3, 19 days). These strong structural

and biochemical differences between the GalC7 hydrogel and the neurospheres conditions may explain the difference of expression of the five markers studied.

As a limitation of this study, we can mention the lack of confirmation of the results at the protein level, considering the experimental difficulties related to the management of this novel hydrogel. Immunocytochemical staining is a very delicate step to do in the case of cells cultured in the supramolecular GalC7 hydrogel. The hydrogel is very soft but also very fragile and is easily removed by the successive steps of the procedure. Since it is not a polymer, there are not any covalent links between molecules. At the end of the culture, if the cells are not embedded in a transparent polymer network before staining, such as polyacrylamide, all the cells will be washed out and they will not be observed. The embedding in polyacrylamide has been performed in a previous publication [20] and requires specific equipment (specific vials, inert atmosphere). For this reason, it cannot be performed routinely. We are currently working on an embedding method that can be implemented more easily.

## 4. Conclusions

This study shows that functional hippocampal cells from mouse embryos (E19.5) can be maintained ex vivo for several days on a supramolecular low molecular weight hydrogel based on a single small molecule, N-heptyl-D-galactonamide (GalC7). New robust conditions have been found, which allow one to reveal different cell characters maintained in culture conditions. It was possible to perform high-quality RNA isolation from GalC7 cultured cells, which opens possibilities to do investigation on transcriptomes via RNA sequencing methods to reveal the differences in transcripts of genes or gene expression differences between different tissues. Embryonic hippocampal cells cultured on GalC7 gel had a high level of expression of the transcription factors *Sox8*, *Sox9*, and *Sox10*. The high level of *Sox10* marker tended to show a higher development of oligodendrocytes. The GalC7 hydrogel is also interesting for neuron growth. The level of expression of Neurod1, a marker of neuronal differentiation, has an interesting level for cells grown for 7 days on GalC7 and in neurospheres that have been grown secondary in GalC7 hydrogels. This level was around 1/4 of the level observed in fresh hippocampi, highlighting neuronal differentiation in these conditions. It has been shown that small molecules, but also grooves can help the reprogramming of somatic cells [36,66]. The low molecular weight supramolecular hydrogel GalC7, acting as a new molecule and scaffold, provides both chemical and physical cues that deserves to be explored further. They provide conditions that are not obtained with any other polymer scaffolds (natural or synthetic) or with scaffold-free culture conditions. Detailed analysis of gene expression also gives a very useful quantitative insight of the cell differentiation distribution on these hydrogels compared to conditions studied in other works. They can be quite easily analyzed because the gel can be easily dissolved. Several other directions, such as changing slightly the molecular structure or the architecture of the hydrogel, notably the carbohydrate head, studying its metabolization, its use for cell induction, or studying the combination with other neuronal factors, would be interesting to explore in further studies.

**Author Contributions:** Conceptualization, A.B., Y.Ö. and M.R.; methodology K.K.B.; software K.K.B., A.B.; validation K.K.B.; formal analysis A.B.; investigation K.K.B., Z.Y. and E.M.; resources J.F., M.R. and Y.Ö.; data curation, K.K.B. and J.F.; writing—original draft preparation, K.K.B., J.F. and M.R.; writing—review and editing A.B, J.F. and M.R., visualization, K.K.B., J.F. and A.B.; supervision, M.R.; project administration, K.K.B. and M.R.; funding acquisition, J.F. and M.R. All authors have read and agreed to the published version of the manuscript.

**Funding:** This work is supported by the grant 2019–2020 of La Fondation Nestlé France to Minoo Rassoulzadegan, The Scientific Research Projects Unit at Erciyes University, by the French National Research Agency (grant N°ANR-15-CE07-0007-01). The European Union is also acknowledged for its financial support for equipment (FEDER-35477: "Nano-objets pour la biotechnologie").

**Institutional Review Board Statement:** The study was conducted according to the guidelines of the Declaration of Helsinki and approved by the Ethics Committee of Erciyes University (protocol code 16/132 and date of approval 16/11/2016).

**Informed Consent Statement:** Not applicable.

**Data Availability Statement:** Data is available on demand to the corresponding author.

**Acknowledgments:** Not applicable.

**Conflicts of Interest:** The authors declare no conflict of interest.

## Abbreviations

| | |
|---|---|
| CNS | Central nervous system |
| Dcx | Doublecortin |
| DMEM/F12 | Dulbecco's modified Eagle medium/nutrient mixture F-12 |
| ECM | Extracellular matrix |
| GalC | Galactocerebrosides |
| GalC7 | N-heptyl-D-galactonamide |
| HBS | Hanks' balanced salt |
| hNSC | Human neural stem cell |
| ITS | Insulin-Transferrin-Selenium-A |
| NFW | Nuclease-free water |
| PBS | Phosphate-buffered saline |
| pHEMA | Poly(hydroxyethylmethacrylate) |
| SAFIN | Self-assembling fibrillar network |
| SDS | Sodium dodecyl sulfate |
| Sox10 | SRY-box 10 |
| Sox8 | SRY-box 8 |
| Sox9 | SRY-box 9 |
| SRY | Sex Determining Region Y |

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
