# Peer review of "Gene Expression of Mouse Hippocampal Stem Cells Grown in a Galactose-Derived Molecular Gel Compared to In Vivo and Neurospheres"

_processes, doi:10.3390/pr9040716_

Round 1
Reviewer 1 Report
Dear authors,
Please find below my comments concerning your manuscript entitled « Gene expression of mouse hippocampal stem cells grown in a galactose-derived molecular gel compared to in vivo and neurospheres » you submitted for publication to Processes.
Korkmaz Bayram et al. describe here a study that aims at determining in vitro appropriate experimental conditions to induce neuronal cell differentiation. For this, they report a comparative analysis of the expression profile of five genes (Sox8, Sox9, Sox10, Dcx, and Neurod1) presented by hippocampal-derived cells cultured under three distinct conditions, namely in neurospheres for 19 days, in N-heptyl-D-galactonamide (GalC7) gel for 7 days, and in both neurospheres and GalC7 gel. The authors show that GalC7 induces a differentiation and maturation of neural progenitor cells superior to those observed with polymer-based scaffolds and conventional culture conditions.
The rationale of this study is interesting and the experimental design described appears to be adequate with the expressed biological question. The methodological approaches are well described. Nevertheless, we can regret the absence of an immunocytochemical exploration that would bring additional elements of interest, notably in terms of cell morphology and spatial organization of cell subtypes in primary cultures. Another limit of the work resides in the absence of biological replicates to attest the relevance of the data presented. Despite these shortcomings, the results are clearly presented and already provide informative findings on the impact of the three conditions tested on the differentiation capacity of the hippocampal-derived cells. On the form, the manuscript suffers from a number of approximations that should be corrected.
Specific comments:
The main limitation of the present work is that the qPCR data is not complemented by a visualization of the primary cultures generated in the different experimental conditions. Such an immunocytochemical approach would have the interest to provide important information on the number of positive cells for the different targets tested but also to produce data on the distribution of the distinct cell types but also their morphology.
Minor points:
In the Introduction section, add references related to Doublecortin and Neurod1 markers in lines 140 & 144, respectively.
In the Material and Methods section, start with the parts "Mice", "Dissection of embryonic hippocampus" and "Gender detection of E19.5 embryos".
- Lines 196-200: Need to clarify
- Table 1: Primers must be written on a single line; Replace 0,01 mM by 10 mM
Figures
- In Figure 1, add the numbers indicated in lines 97-99 in the arrows starting from Hippocampus at 19.5 pc and modify the title by replacing "four conditions for the cell culture of mouse embryo hippocampi primary cells followed by gene 114 expression analysis by qPCR” by " four conditions for the primary cell culture of mouse embryo hippocampi-derived cells followed by qPCR analysis of gene expression”.
In figure 4, replace "The absorbance value of GalC7 gel without cultured neuronal cells (a) and with cultured neuronal cells (b)." by "The absorbance value of GalC7 gel without (a) and with (b) cultured neuronal cells."
In figure 5, replace, by .. What do the numbers after GalC7, Neuropsph D19 and Neurosph D12 + GalC7 D7 correspond to? Technical replicates?
On the form, the quality of the submitted article needs to be improved.
In the document, it is necessary to replace l by L to give the volume in liter and to homogenize rotation speeds in g or rpm.
- Line 47: "all aspects of cells complexities" should be replaced by "all aspects of cell complexities”.
- Line 58: Add (CNS) after central nervous system.
- Line 79: Delete a space after the reference 15
- Lines 127 & 128: Delete the "S" in astrocytes precursors and oligodendrocytes precursors.
- Line 146: Replace "central nervous system" by abbreviation.
- Line 147: Delete a point after the reference 56.
- Line 391: Suppress N- before GalC7
Overall, the manuscript presented here provides new data on the in vitro modalities to be applied to allow an advanced differentiation of neuronal cells and thus to approach their in vivo behavior. In regard of its interest in the central nervous system development, I consider that the manuscript could be recommended for publication if the authors are able to answer the point raised above concerning the expression of these markers by the cells in vitro.
Author Response
First, we would like to thank Editor and the Reviewers for their time and efforts spent on the close reading and for the proper suggestions and the advices which, we think, have enhanced the quality manuscript. We agree with reviewers and have carefully gone through with reviewers’ comments and suggestions and made required modification. The present version of the paper has been revised according to the reviewer’s suggestions, below you can find our answers and additional comments to reviewers.
Reviewer 1:
Specific comments:
The main limitation of the present work is that the qPCR data is not complemented by a visualization of the primary cultures generated in the different experimental conditions. Such an immunocytochemical approach would have the interest to provide important information on the number of positive cells for the different targets tested but also to produce data on the distribution of the distinct cell types but also their morphology.
We would like to thank reviewer for his/her fair comment, ‘immunocytochemical staining on this new material’ is one of the ongoing research topics. Immunocytochemical staining is a very delicate step to do in the case of cells cultured is the supramolecular GalC7 hydrogel. Actually, the hydrogel is very soft but also very fragile and is easily removed by the successive steps of the procedure. Because it is not a polymer, there is not any covalent links between molecules. At the end of the culture, if the cells are not embedded in a transparent polymer network such as polyacrylamide before staining, all the cells are washed out and they cannot be observed. The embedding in polyacrylamide has been performed in a previous publication by the chemists of the team and requires specific equipment (specific vials, inert atmosphere). For this reason, it cannot be performed routinely. We are currently working on an embedding method that can be implemented more easily. This better solution will be published in following research papers.
Minor points:
In the Introduction section, add references related to Doublecortin and Neurod1 markers in lines 140 & 144, respectively. – Reference added
In the Material and Methods section, start with the parts "Mice", "Dissection of embryonic hippocampus" and "Gender detection of E19.5 embryos". – Order of paragraphs modified as suggested
- Lines 196-200: Need to clarify – Required clarification added to paragraph
- Table 1: Primers must be written on a single line; Replace 0,01 mM by 10 mM: - Table 1 modified
Figures
- In Figure 1, add the numbers indicated in lines 97-99 in the arrows starting from Hippocampus at 19.5 pc and modify the title by replacing "four conditions for the cell culture of mouse embryo hippocampi primary cells followed by gene 114 expression analysis by qPCR” by " four conditions for the primary cell culture of mouse embryo hippocampi-derived cells followed by qPCR analysis of gene expression”. -Done
In figure 4, replace "The absorbance value of GalC7 gel without cultured neuronal cells (a) and with cultured neuronal cells (b)." by "The absorbance value of GalC7 gel without (a) and with (b) cultured neuronal cells." -Done
In figure 5, replace, by .. What do the numbers after GalC7, Neuropsph D19 and Neurosph D12 + GalC7 D7 correspond to? Technical replicates?
1, 2, 3, 4 refer to the type of experiment, as explained on Figure 1 and on lines 97-99. To make it clearer, we indicated those numbers into brackets. D19, D12 and D7 represent the number of days of culture, clarification added to figure legend.
On the form, the quality of the submitted article needs to be improved.
In the document, it is necessary to replace l by L to give the volume in liter and to homogenize rotation speeds in g or rpm. -Done
- Line 47: "all aspects of cells complexities" should be replaced by "all aspects of cell complexities”. -Done
- Line 58: Add (CNS) after central nervous system. -Done
- Line 79: Delete a space after the reference 15 -Done
- Lines 127 & 128: Delete the "S" in astrocytes precursors and oligodendrocytes precursors. -Corrected
- Line 146: Replace "central nervous system" by abbreviation. -Done
- Line 147: Delete a point after the reference 56. -Corrected
- Line 391: Suppress N- before GalC7 -Deleted
Reviewer 2 Report
The authors analyzed the differentiation of mouse hippocampal stem cells under four culture conditions by characterization of several gene expression profiles by qRT-PCR. The manuscript is not well presented. The “Instruction” section is way too long. The “Material and Methods” section contains too many experiment details that can be found from the manufacturer’s instructions. The manuscript presents very limited information and fails to advance our understanding of stem cell differentiation. Further, this manuscript needs intensive English editing. Therefore, I do not recommend the publication of this manuscript.
Reviewer 3 Report
This manuscript is a beautiful study by Bayram et al, where the authors studied gene expression of mouse hippocampal stem cells grown in a 3D ex vivo culture. The impact of this study would be very high. I commend the authors for their extensive efforts. However, one critical issue that still prevails is that the authors have only looked at a few cherry-picked genes by qPCR to compare their systems. I would prefer seeing RNA-seq analysis between samples to compare their transcriptome properties unbiasedly. If the other reviews are favorable, I would at least like the authors comment about this in the discussion section. Otherwise, this article would be a nice addition to MDPI-Processes.Author Response
Please see the attachment.

Reviewer 4 Report
The present paper gives a great conditions for the differentiation and maturation of neural progenitor cells. They show that the GaIC7 hydrogel helps researchers to start new procedures and create new chances for tissue engineering and transcript analysis.
Reviewer 5 Report
The article ‘’Gene expression of mouse hippocampal stem cells grown in a galactose-derived molecular gel compared to in vivo and neuro-spheres’’ is a well-written article by Bayram et al. I would like to suggest few comments below.
1.Mice with vaginal plugs were followed up for 0.5 days of pregnancy was mentioned in the paper. The authors should correct this sentence.
2.In figure 5, the authors should include the p Values.
3.The authors claim that GalC7 hydrogel is favorable in vitro condition for neurogenesis and the differentiation of the primary cells into neurons. Hence, the authors should also analyze the differentiation-inducing markers like Notch1, MASH1(ASCL1), FOXA2, NURR1
4.The authors should also include visual Immunofluorescence staining like F-ACTIN, vinculin or paxillin of cells on the GalC7 hydrogel.
Round 2
Reviewer 1 Report
Dear authors,
I think the revised version has become clearer and more pleasant to read. The majority of the points I raised have been considered.
I appreciated your comment in the cover letter about the lack of immunostaining being the main limitation of the work presented here. I think it would be appropriate to add a sentence in the discussion to mention in the opening that the confirmation of the results at the protein level is in progress taking into account the experimental difficulties related to the management of the hydrogel.
Upon rereading, I noticed a few typos.
- line 280 and 411, replace 4000 and 1000 by 4,000 and 1,000 respectively
- line 303 (Table 1), replace 10 mM with 10 uM
- Throughout the manuscript, replace l with L for liter
Reviewer 2 Report
The authors addressed most of my concerns. Thus, the manuscript could be accepted for publication.
Author Response
We thank reviewer for comments.